# Enterprise Resource Planning Systems: Digitization of Healthcare Service Quality

**Muhammad Fiaz [1], Amir Ikram [2,3,*] and Asad Ilyas [1]**

[1]   Institute of Business & Management, University of Engineering & Technology, Punjab 54890, Pakistan; muhammadfiaz@mail.xjtu.edu.cn (M.F.); asad.ilyas@live.com (A.I.)
[2]   School of Management, Xi'an Jiaotong University, Xi'an 710049, China
[3]   Department of Business Management, National College of Business Administration & Economics, Lahore 54660, Pakistan
*   Correspondence: amirikram@stu.xjtu.edu.cn

**Abstract:** The purpose of this study is to evaluate the perception of healthcare professionals in improving the quality of services in healthcare centers by deploying the platform of Enterprise Resource Planning (ERP). Individual attributes, organizational impression, information, and the system quality of ERP have been used to evaluate the overall influence of integrated planning systems on health care service quality. A mixed methods approach is used to collect and examine data through triangulation. Data for the empirical study was collected from 279 medical professionals of five healthcare organizations operating in the city of Lahore, Pakistan, through a self-administered questionnaire. Descriptive statistics squared multiple correlations and reliability coefficients were used as data analysis tools. Moreover, the goodness of fit test of the structural model was conducted through AMOS 20. All given dimensions of ERP are postulated to have a positive effect on healthcare service quality. The results reveal that the use of an enterprise planning system has a positive impact on individuals, organizational information quality, and system quality in healthcare services. The study further concludes that a well implemented 'Enterprise Resource Planning System' results in better system output and enables healthcare professionals to provide better healthcare service quality.

**Keywords:** enterprise resource planning; information systems; healthcare; service quality

## 1. Introduction

Database management systems and information analytics ensued widespread improvements of quality and efficiency in almost every sphere of life. Moreover, the healthcare sector is no exception, where technological advancements and information systems are transforming the sector with the orientation of integrated systems. Most of the transformations about healthcare systems are primarily focused on health care quality and the minimization of cost. Healthcare professionals and institutions lack the adequate systems required to deliver strategic change. Thus, there is a sense of urgency on their part to make use of information technology (Cucciniello et al. 2016). However, the adoption rate of the IT-based integrated system in the healthcare sector is quite slow as compared to other sectors, such as commerce and finance, especially in the case of emerging economies. This calls for an empirical investigation of the healthcare sector concerning problems being faced in the implementation of the Enterprise Resource Planning (ERP) system. ERP software can integrate all the departments of an organization and functions into a unified system (Somers and Nelson 2001). ERP primarily offers two welfares to the organization that are usually not prevalent in an organization with a non-integrated system; i.e., (a) Unified database for ERP systems where all the transactions of an organization are entered, stored, handled, and reported. (b) A single enterprise business view that

covers all departments of an organization and its functions. Achieving a competitive edge is considered another reason for implementing ERP systems in healthcare organizations (Al-Mashari et al. 2003).

The design of ERP systems is based on the advancement of the ability of an organization to generate accurate and timely information, thereby enhancing the competitiveness of an organization (Singla 2008). These systems are complex and incur the high cost of implementation in an organization. Therefore, organizations need to reconsider their ERP system purchasing and implementation plan (Razi and Tarn 2015). The ERP system has substantial benefits, but it can also be responsible for causing troublesome changes to an organization and the rate of failure of the ERP system varies. Healthcare is a very complex sector comprising a large number of departments and patient care systems (Acharyulu 2012). Studies have advocated that there exist a positive relation of investment in an information system with health care service productivity (Menon et al. 1996). The reliability of the healthcare system is increasing due to the implementation of an information system for the help in the diagnosis of disease, improved management, and better services (Adler-Milstein and Bates 2010). Information system (IS) in healthcare is explained as an enormous system of integration that has the capability to support the colossal information need of the healthcare system covering the financial, clinical, auxiliary, and patient management (Roussel et al. 2006). Implementation of IS in healthcare has resulted in enhanced patient care quality, improving the management of health services, and access to knowledge for decision making in administration and clinical processes (Parr and Shanks 2000). Integral systems are requirements of healthcare for the procedure development and application of patients at the same time considering the requirement of capacity. In the healthcare system, the ERP system use has improved the material management process.

There is hesitation on the part of practitioners in accepting new technologies, especially in the public sector of developing economies such as Pakistan. Many rely on persisting with the traditional way of doing things, and that is truer for the public healthcare sector. However, the private sector in developing countries is relatively technology privy because of their commercialized nature. ERP is widely implemented in private sector, and there are few queries of its implementation in public healthcare, particularly in emerging economies. Hence, this paper aims to enrich the debate concerning public healthcare by investigating one of the first applications of it in the emerging economy of Pakistan. The rest of the article is structured as follows: the next section reviews the literature on enterprise resource planning and its implementation in the public healthcare sector with specific reference to developing economies. Following that, the theoretical model and hypotheses development are discussed. In the next section, the research methodology and framework is elaborated upon in the research design section. Conclusions, research implications, and provocative questions for future study bring the discussion to a close.

## 2. Literature Review

Healthcare service quality is the result of a collaboration between the healthcare agent and the patient. Thus, personal characteristics of the provider and the patient and organizational factors affect the overall service quality (Mosadeghrad 2014). Jawaid (2016) analysed recent studies about violent incidents against doctors in South Asian countries and suggested that such incidents are triggered by a lack of communication between healthcare professionals and the patient, reduced image of the medical profession, and below par quality of care. A proper ERP system can take care of these problems. In the medical and healthcare profession, there often occurs a scenario where a doctor lacks knowledge regarding a particular disease, and this is especially common among young doctors. For instance, Al-Arifi et al. (2016) observed that healthcare professionals lack knowledge regarding warfarin interactions with drug and herbal medicines. Another advantage of the ERP system is that it can accommodate for the lack of information regarding a specific disease or medicine on the part of doctors. Bharati and Ganguly (2013) noted that the South East Asian region has the highest number of malaria cases. They suggested that increased monitoring, surveillance, and cross-border collaboration can alleviate this problem. Balkrishnan et al. (2013) proposed that active Comparative Effectiveness

Research (CER) systems nurture the sharing of resources, skills, and capabilities. ERPs can be helpful in this regard as they can enable regional networking and data-sharing among all stakeholders in real time.

Rouhani and Mehri (2018) investigated ERP benefits through a survey by defining 31 empowering benefits for this enterprise system based on reviewing the literature and classifying it into four groups of empowering benefits including informative, communicative, growth and learning, and strategic benefits. The results indicated that the communicative, strategic and informative empowering benefits are significant advantages. Tasevska et al. (2014) conducted a survey on 30 SMEs in the Republic of Macedonia. The findings revealed that SMEs implemented common project planning practices, although they did not consider the planning process as a distinct phase of the ERP implementation. Considering the success of the ERP implementations, this study established that most of the representatives perceive the undertaking as useful regarding client satisfaction and perceived quality measures. By using a multi-method approach, Cucciniello et al. (2016) conducted comparative case studies of two different hospitals to examine the adoption and implementation of identical medical record systems and found that healthcare organizations benefited from deploying the integrated medical system regarding information quality, data sharing, and cost efficiency. The ERP system has the capability of integrating information that is used by human resources, manufacturing, distribution, and accounting departments into a single computer system (Umble et al. 2003). ERPs provide a holistic view to organizations about every business process ongoing in the organization. It provides one software application, a unified database, and a single interface to the organization. The ERP system can provide enhancement in service quality, productivity, service cost decrease, and efficiency (Shaul and Tauber 2013). The earlier target of the ERP system was not the services sector, but instead, the ERP system vendors focused on the manufacturing companies (Botta-Genoulaz and Millet 2006).

The Enterprise Resource Planning system emerged in 1960, beginning with material requirement planning (MRP). After that there was an advancement in the system and it was transformed into MRP II (Basoglu et al. 2007). From 1960 to 1970, there was a paradigm shift from inventory control to MRP improvement, and this was accepted by a lot of manufacturing companies for the efficient calculation of the materials they required for the manufacturing process. Then the MRP system further evolved into a system that was more sophisticated and included detailed capacity planning, master scheduling, long-range planning, capacity planning, and resource planning (Xue et al. 2005). MRP II systems further included planning related to operations and sales, as well as a financial interface. The MRP II system was a useful tool for planning for all types of resources present in an organization. It was logical for the planning of materials and production processes, but organizations realized the need for the incorporation of customer satisfaction and profitability (Wallace and Kremzar 2002). The latest form of the ERP system present today is capable of handling with various business units covering management of customer relationship, human resource, purchasing, finance and accounting, manufacturing, processing of the order, management of materials and planning of operation and sales (Botta-Genoulaz and Millet 2006). As a result, a large number of organizations have already adopted ERPs, and its implementation is fast increasing in the services sector (Acharyulu 2012). The services sector has dominated its share in the Growth Domestic Product (GDP) of developed countries. Therefore, the progress of technology, globalization trends, and communication technologies have exerted pressure on the service sector for their new competition offerings (Costa 2015). The organizations are interested in information system acquiring for the adoption of modern technology and making it accessible for the users. The adoption of information technology in the healthcare system is complicated as the Enterprise Resource Planning system in healthcare is concerned with the lives of humans (Bazhair and Sandhu 2015). The acquiring of this system is further influenced by various actors that have multiple interests and backgrounds. There are chances that stakeholders in health care will resist in the adoption of information system and there is need that it should be managed carefully, as these actors will be involved in information system adoption and implementation (MacLennan and Van Belle 2014). Various success factors make implementation of

ERP system successful. The perspective includes setting up of an ERP system, implementation phase, evaluation step, success of ERP system and its profit (Botta-Genoulaz and Millet 2006).

Twenty-two critical success factors for ERP system implementation at various implementation phases have been presented by Somers and Nelson (2001). Successful implementations of the ERP system have also been linked with the management of the project, support of top management, teamwork, program of change management, and composition of ERP system team. The culture of an organization, economic environment, and regulation of the government also offer challenges for the implementation of Enterprise Resource Planning system (Ward and Peppard 2016). The practical implementations of an ERP system offers many benefits. The benefits may be intangible or tangible; e.g., improvement in cash flow, order management, system integration, reduction of inventory, logistics management, and information quality enhancement (Botta-Genoulaz and Millet 2006). Some organizations have improved their position by ERP system implementation in their business processes; for example, a company known as Earthgains implemented an ERP system and, as a result, there was an enhancement in their operating margin from 2.4–3.3%. Similarly, on-time delivery was also heightened to 99% (Hong et al. 2012). The ERP system implementation is a very complex process, and organizations encounter different types of problems while adopting ERP system phases (Kumar et al. 2003). In many cases, the implementation of an ERP system has failed because of the enormous implementation costs. The failure rate of implementation of an ERP system has resulted in better ERP system process understanding (Hung et al. 2014). ERP systems have a problem, the very high cost of implementation sometimes provides inadequate results because the individuals using the ERP system are not aware of proper ERP system functioning and working (Altamony et al. 2016). The failure rate of Enterprise Resource Planning system has been greatly publicized, however, it has not detracted organizations from investing money into ERP system implementation (Scotti et al. 2007). The cost of integrating an ERP into the business process system is 3 to 10 times higher than the original cost of the ERP software. This increase is due to the high costs charged by system consultants and the persons involved in system integration (Karimi et al. 2007). As well, there is cost related to the replacement of the existing information system in the organization into the system required for ERP.

There are quite a few studies that analysed the digitization of healthcare service quality with the help of ERP. Almajali et al. (2016) conducted an empirical analysis of 175 Jordanian healthcare organizations and examined the data using structural equation modeling. They found a significant relationship between antecedents of ERP and its implementation success and further suggested that user satisfaction plays a significant mediating role between ease of use and ERP implementation success. Chiarini et al. (2018) investigated ERP implementation in public healthcare and suggested that benefits can be classified into four theoretical categories: patients' satisfaction, stakeholders' satisfaction, operations efficiency, and strategic and performance management. While issues surrounding ERP implementation include the complexity of the project, process re-engineering, and staff involvement. As per Du (2017), the performance of the healthcare service includes quality and efficiency, so there is inevitably an association between them. In general, it is believed that there is a trade-off between quality and efficiency; however, Du (2017) suggested that it can not be completely accurate.

One of the methods of curtailing cost and capitalizing on competitive advantage is the implementation of an ERP system. With respect to developing economies, there is hesitation on the part of practitioners in accepting new technologies and there is a tendency to persist with traditional ways of doing things. Discourse on ERP systems acceptance is prevalent among policymakers and researchers as they intend to understand the underlying psychological and social aspects inducing user adoption behavior. Researchers also seek to answer why the application of ERP among organizational stakeholders remains at a perfunctory level (Lim et al. 2005). Amoako-Gyampah and Salam (2004) proposed an extension to the technology acceptance model and empirically investigated it in an ERP implementation setting. It was shown that both project management and training influence the shared views that users form about the assistances of the technology and how shared philosophies affect the perceived expediency, ease of use, and know-how. There are numerous aspects that affect the

implementation of ERP, including user resistance. Shih and Huang (2009) examined the behavioral intention and actual usage of ERP implementation, grounded in the technology acceptance model. They used the Lisrel package of structural equation modelling to validate the causal associations between variables. Analytical outcomes determine that top leadership support positively affects the perceived efficacy and perceived ease of use. It was also found that behavioral intention positively affects genuine usage.

## 3. Theoretical Model & Hypotheses Development

The impact of the ERP system on healthcare service quality depends on the acceptance of the ERP system in the public sector of developing economies. Thus, implications of the Technology Acceptance Model (TAM) came into play and provide a tangible theory for study.

*Individual impact*: Individual impact refers to an increase in an individual's efficiency by the use of the ERP system in the organization. This research is related to healthcare service quality and includes nurses and doctors and covers four characteristics; i.e., productivity of the individual, learning, decision making in an effective manner, and awareness of the individual. The overall healthcare service quality is dependent on the individual productivity and performance. Use of the ERP system in healthcare plays an eminent role in enhancing the productivity of the individual. Enterprise Resource Planning systems teach individuals with new practices, and therefore enhance learning. Such systems make all data available quickly and have a positive influence on decision making (Kanellou and Spathis 2013). Moreover, the ERP system helps professionals become more aware of their job detail. The following hypotheses are postulated to answer the research objectives that we outlined in the introductory section:

**Hypothesis 1 (H1).** *Individual impact has a positive relationship with service quality in healthcare.*

Organizational impact: Organizational impact refers to the welfares that are received by an organization through the implementation of an ERP system. Impact of an information system is evaluated through business performance in the healthcare sector (Rai et al. 2006). The organizational impact is measured as organizational efficiency, market value, effectiveness, competitive advantage, and strategic value. It includes cost reduction, capacity improvement, business process variation, and productivity enhancement.

**Hypothesis 2 (H2).** *Organizational impact has a positive relationship with service quality in healthcare.*

*Information quality:* Information quality deals with the quality of the information system. Information quality dimensions include accuracy, currency, completeness, and consistency (Acharyulu 2012). ERP systems are valuable in information sorting but are not as good for providing information to a user about specific benefit.

**Hypothesis 3 (H3).** *Information quality impact has a positive relationship with service quality in the healthcare sector.*

*System Quality:* System quality refers to the extent to which the information system is evaluated concerning the technical and design aspects. According to Gorla et al. (2010), it is the measurement of the soundness of the system. Delone and McLean (2003) mention the attributes of system quality measurement; i.e., quality of data, the reliability of data, ease of use, functionality, and integration. System quality covers the facts that the system is easy to use, the quality of documentation, the presence or absence of bugs in system, and the maintenance of quality of the program (Cucciniello et al. 2016).

**Hypothesis 4 (H4).** *System quality impact has a positive relationship with service quality in healthcare.*

By the abovementioned formulated hypotheses, a research model is proposed to evaluate the impact of individual, organizational, information, and the system quality of an ERP on healthcare service quality (Figure 1).

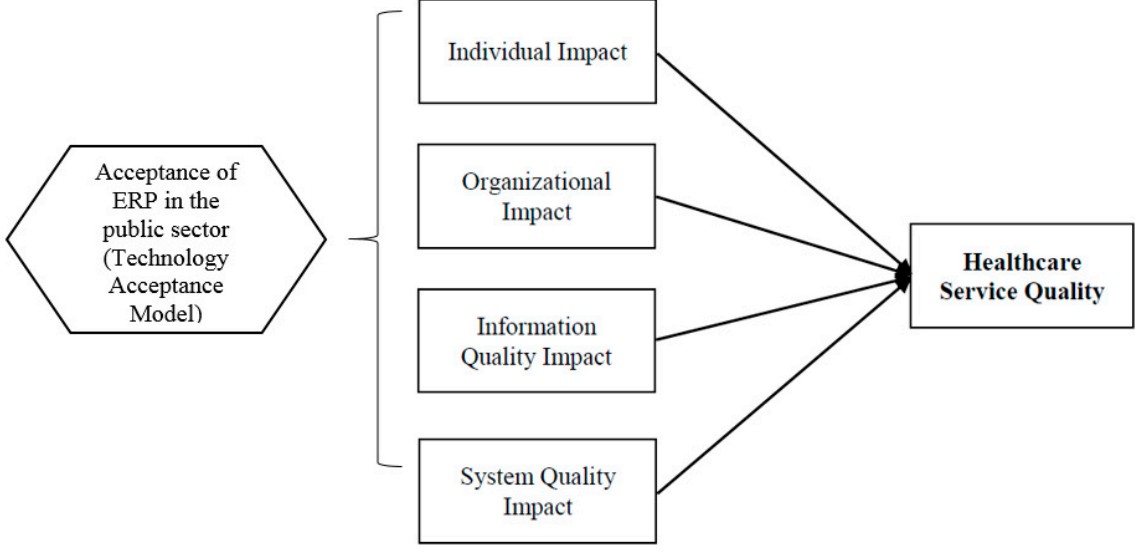

**Figure 1.** The proposed research model.

## 4. Research Design

*Design of Questionnaire:* The survey instrument of Questionnaire is utilized for data collection. The questionnaire has been reproduced from the service quality of healthcare and information system dimensions. The research instrument is based on three parts. The first part covers the use and impact of ERP Systems; the second part deals with the service quality of healthcare, and the third part is comprised of the demographic variables.

The first part of the questionnaire is based on five impact variables, namely, individual impact, system quality impact, organizational impact, information quality impact, and overall impact by the use of ERP in a healthcare setting, which was analysed through questions as put forward by Gable et al. (2008).

System quality impact describes the performance grade of the ERP from a technical perspective. It covers system accuracy, customization and integration, ease of use, efficiency, system features, ease of learning, the sophistication of system, access, reliability, and flexibility and user requirements. Information quality impact deals with the quality of the information created by the ERP system to be used in a healthcare setting. This variable is measured with the help of ten questions, which includes content accuracy of the medical field, the usability of information for medical staff, conciseness, availability of information, timeliness and uniqueness, importance, relevance, format, and comprehensibility. Organizational impact is measured by using eight questions that deal with improved outcomes, organizational cost, e-government, cost reduction, business process change, staff requirement, increased capacity, and overall productivity. The individual impact is analysed with the help of four questions that cover individual productivity, awareness, and knowledge of doctors about their field, individual learning, and decision effectiveness. Overall, the variable 'impact' covers questions about overall ERP system including impacts on the individual and impact on the organization.

The second section of questionnaire deals with the performance of service quality in healthcare. The measurement of this variable can be made by measuring Responsiveness, Reliability, Assurance, and Empathy. The respondents had to answer seventeen questions, and these were reproduced using the SERVQUAL scale, following the dimensions of healthcare setting as reported by Lee et al. (2000).

These questions were close-ended with multiple choices on the five-point Likert scale. Closed questionnaires are easy to fill out by respondents and also suitable for analysis. The Likert scale in the present study ranged from 1 to 5. 1 = strongly disagree, 2 = disagree, 3 = neither disagree nor agree, 4 = agree, and 5 = strongly disagree. The last part deals with the demographic information of the respondent (medical staff) and includes name, position in the organization, and relevant experience.

*Collection of data:* The study uses a mixed methods approach, wherein both primary and secondary sources are utilized for the collection of data. For the empirical investigation, questionnaires were used for the collection of primary data, whereas archival data is used to collect secondary data for model construction. In total, 500 self-administered questionnaires were disbursed to respondents of five healthcare organizations in the city of Lahore, Pakistan[1]. We selected both the private and public healthcare organizations, as the primary consideration was the implementation of ERP system. Out of which, we got feedback from 279 respondents, and this depicts that the response rate is 56%. The respondent's demographic characteristics are presented in Table 1.

**Table 1.** Demographic characteristics of the respondents.

| Measure | Value | Frequency | Percentage |
|---|---|---|---|
| Gender | Male | 156 | 55.9% |
|  | Female | 123 | 44.1% |
| Age | <25 | 104 | 37.3% |
|  | 26–30 | 95 | 34.1% |
|  | 31–35 | 57 | 20.4% |
|  | 36–40 | 13 | 4.7% |
|  | >40 | 10 | 3.6% |
| Degree of Experience | <3 | 92 | 33.0% |
|  | 3–5 | 114 | 40.9% |
|  | >5 | 73 | 26.2% |
| Job Title | Doctor | 60 | 21.5% |
|  | Medical Staff | 149 | 53.4% |
|  | Administration Staff | 70 | 25.1% |

## 5. Data Analysis

Table 2 presents the descriptive statistics, reliability coefficient, and inter-item correlations between the variables. The results of the descriptive statistics for this study demonstrate that the mean of the overall 'ERPs impact' construct is greater than 3. Thus, it is concluded that medical professionals believe that the use of ERPs has positive individual, organizational, information quality, and system quality impact. The value of all reliability coefficients ranges between 0.74 to 0.86, which is well within the threshold level as endorsed by Revelle (2014).

**Table 2.** Descriptive, reliability coefficients & correlation matrix.

| Construct | Mean | Standard Deviation | Reliability ($\alpha$) | 1 | 2 | 3 | 4 |
|---|---|---|---|---|---|---|---|
| Individual Impact | 3.78 | 0.59 | 0.74 | 1 |  |  |  |
| Organizational Impact | 3.60 | 0.47 | 0.76 | 0.566 ** | 1 |  |  |
| Information Quality Impact | 3.32 | 0.38 | 0.83 | 0.224 ** | 0.306 ** | 1 |  |
| System Quality Impact | 3.59 | 0.57 | 0.85 | 0.444 ** | 0.405 ** | 0.309 ** | 1 |
| Healthcare Service Quality | 3.62 | 0.51 | 0.86 | 0.518 ** | 0.546 ** | 0.416 ** | 0.666 ** |

** Correlation is significant at the 0.01 level (2-tailed).

---

[1] The five focal healthcare organizations for data collection are as follows: Doctors Hospital, Shaukat Khanum Memorial Cancer Hospital & Research Centre, Surgimed Hospital, Shalamar Hospital, and Shaikh Zayed Hospital.

Correlation matrix for dependent and independent variables reveals that individual impact (independent variable) is significantly correlated ($r = 0.518$, $p < 0.01$) with service quality in healthcare (dependent variable). Likewise, the organizational impact is positively correlated ($r = 0.546$, $p < 0.01$) with service quality in healthcare. Information quality impact also has a positive correlation ($r = 0.416$, $p < 0.01$) with service quality in healthcare. "System quality impact" is found to be positively correlated ($r = 0.666$, $p < 0.01$) with service quality in healthcare. Therefore, correlation of independent variables including individual, organizational, and system quality impact is strong in nature and the correlation of "Information Quality Impact" with "Healthcare Service Quality" is a moderator in nature as described by Field (2013).

*The Structural Model:* The research model is tested for goodness of fit by using AMOS 20.0 and the results suggest that the overall model can be categorized as robust (Figure 2). The fit indices of the structural model along with the recommended values are also shown in Table 3. The first Goodness of Fit measure is the ratio between chi-square ($\chi^2 = 36.708$) and degree of freedom (df = 13). The threshold value for this measure is $\leq 3$ (Carmines and McIver 1981), and in this research, it is suitable as a computed value for this research is 2.82369. The values of the Tucker–Lewis index (TLI 0r NNFI) and Comparative Fit Index (CFI) are all above 0.90, signifying a good model fit as per the standard set down by Kline (2015). Similarly, the value of the Normed Fit Index (NFI) is also higher than 0.90, which is considered a good fit measure of the model (Murtagh and Heck 2012). The value of RMSEA is also below the significant value. All of these fit indices are acceptable as per the criteria laid down by different researchers, suggesting a functional model fit for this data.

**Table 3.** Fit indices of structural model.

| Goodness-of-Fit (GOF) Measure | Score | Threshold Value |
|---|---|---|
| $\chi^2$ | 36.708 | |
| Df | 13 | |
| $\chi^2/\text{df}$ | 2.8239 | <=3 |
| Normed fit index (NFI) | 0.955 | >0.90 |
| Tucker-Lewis index (TLI or NNFI) | 0.934 | >0.90 |
| Comparative Fit Index (CFI) | 0.97 | >0.90 |
| RMSEA | 0.079 | <0.080 |

Table 4 presents the path analysis for the study variables as individual impact has a significant positive effect on healthcare service quality ($\beta = 0.136$, $p < 0.001$), so hypothesis 1 is supported. The organizational impact has a significant positive effect on healthcare service quality ($\beta = 0.296$, $p < 0.001$) similarly information quality impact also has the significant positive effect of healthcare service quality ($\beta = 0.235$, $p < 0.001$), so hypothesis 3 is also supported. Like previous hypotheses, system quality impact has a positive effect on healthcare service quality, which is also statistically significant ($\beta = 0.485$, $p < 0.001$), so hypothesis 4 is also supported. Figure 2 shows the estimates and the value of squared multiple correlations (SMC), which are similar to the value of the R square in the regression analysis. As the value of SMC is 0.575, this shows that 57.5% of the variation in 'Healthcare Service Quality' is explained by all the independent variables.

**Table 4.** Results of hypotheses testing.

| Hypotheses | Estimate | Results |
|---|---|---|
| H1: Individual impact has a positive relationship with service quality in healthcare. | 0.136 *** | Supported |
| H2: Organizational impact has a positive relationship with service quality in healthcare. | 0.296 *** | Supported |
| H3: Information quality impact has a positive relationship with service quality in healthcare. | 0.235 *** | Supported |
| H4: System quality impact has a positive relationship with service quality in healthcare. | 0.485 *** | Supported |

*** $p < 0.01$.

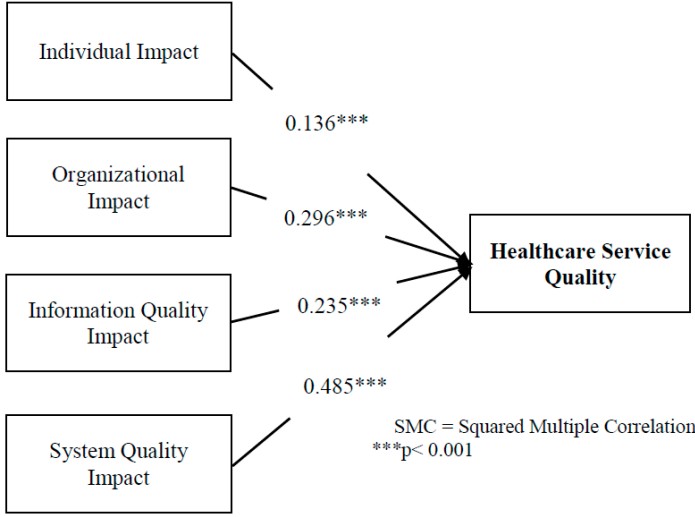

**Figure 2.** Structural model results.

## 6. Conclusions

The purpose of this article is to study the effect of ERP implementation success antecedents (such as information quality impact, system quality impact or organizational impact) on ERP implementation success itself. The understanding of factors affecting health service quality helps in benchmarking the practice and reduction of the errors rate in the sensitive medical profession. We investigated the impact of ERP on the quality of services in healthcare and revealed a positive result of ERP implementation in healthcare. ERP had a positive impact on the professional work of the medical professionals examined in our study. Additionally, the ERP system helped them with their organization. It has also been suggested that the incorporation of ERP systems in healthcare organizations is a major source of achieving a competitive edge, especially in the private sector. To expedite the adoption of ERP based systems, an endorsement of the entire medical staff should be ensured by the pursuance of a decentralized decision making process and the participation of clinical staff. In a recent study, Cucciniello et al. (2016) observed that strong backing from clinical staff and involvement among professionals (clinicians and nurses) is imperative for the adoption and implementation of the integrated medical system. Healthcare service providers should monitor healthcare quality on a regular basis and accordingly initiate continuous quality improvement programs, such as ERP. Researchers interested in evaluating the ERP system's success in a healthcare setting can utilize the proposed model.

The novelty and value of this paper lie in a new classification of benefits and criticalities concerning enterprise resource planning implementation in public healthcare. Previous studies (Al-Mashari et al. 2003; Razi and Tarn 2015; Lim and Loosemore 2017) mostly focused on the technical aspects of the ERP system, and usually did not differentiate between the public and private sector. This research contributes to knowledge by revealing the fact that a well-applied ERP system in a public organization will produce a better output to the system, and also assist medical professionals in providing better service quality in the healthcare sector. Five healthcare public sector organizations were part of this study. The paper also contributes to healthcare practice by developing a conceptual framework that provides policy-makers with a practical understanding of factors that affect healthcare service quality. The impact dimensions of system quality impact, individual impact, information quality impact, and organizational impact and their relation to service quality in healthcare have been presented. The results show that public healthcare institutions are also receiving many benefits after the adoption of ERP. Operational benefits were mentioned by all the institutes involved. The most important benefits were report generation, quick access to critical information, a better product,

and cost planning. Respondents also mentioned that they had reduced corruption as a result of ERP implementation. It is also a substantial benefit considering the high corruption rate in Pakistan. Along with the benefits and challenges faced by Pakistani enterprises, infrastructure problems like electricity, difficulties with the integration of one module with other module, the costs of adoption, and lack of skilled ERP consultants must be included. Additional studies in this regard are recommended on cloud-based ERP, which is fast growing all around the world. Moreover, respondents were selected from healthcare organizations of a metropolitan city of Pakistan, and thus the results cannot be generalized to other countries or developed countries. Another limitation of the study originates from the fact that the field research was conducted in the public sector hospitals of Lahore city, which is the second largest city in Pakistan. There are limitations regarding the research design. The sample size was fairly small and it is suggested to replicate the study on a larger scale, and maybe extend it to other industries.

**Author Contributions:** "Conceptualization, M.F. and A.I. (Asad Ilyas); Methodology, A.I. (Amir Ikram); Software, A.I. (Asad Ilyas); Validation, A.I. (Amir Ikram), A.I. (Asad Ilyas) and M.F.; Formal Analysis, M.F.; Investigation, A.I. (Amir Ikram); Resources, A.I. (Asad Ilyas); Writing-Original Draft Preparation, M.F., A.I. (Asad Ilyas), and A.I. (Amir Ikram); Writing-Review & Editing, A.I. (Amir Ikram).

**Funding:** This research was funded by 'Humanities and Social Sciences of Ministry of Education Planning Fund [13YJA630078]'.

**Conflicts of Interest:** The authors declare no conflict of interest.

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
