# Peer review of "Enterprise Resource Planning Systems: Digitization of Healthcare Service Quality"

_admsci, doi:10.3390/admsci8030038_

Round 1
Reviewer 1 Report
The purpose of this paper is to investigate the perception of healthcare professionals in improving the quality of services in healthcare centers by deploying the platform of ERP system through the sample data of 279 medical professionals, collected from five healthcare organizations operating in the city of Lahore, Pakistan. The research results indicate that a well implemented ‘Enterprise Resource Planning System’ results in better system output and enable healthcare professionals to provide better healthcare service quality. This is an interesting paper and can be accepted for publication after the following revisions:
Suggest enhancing “Theoretical Model & Hypotheses Development” section. The literature review is not enough before proposing a hypothesis.
In my opinion, the research model proposed in Figure 1 is based on the updated Information Success Model proposed by DeLone and McClean (2003). However, this paper did not discuss the relationship (or the similarities and differences) between this paper' model and DeLone and McLean’s model.
Suggest adding the measurement of variables in this research.
Suggest enhancing the description of the population of this research and whether the sample of 279 medical professionals can be representative of the population.
Suggest citing the related papers published in “administrative sciences” (no one now) to link this paper to the material of the journal. Otherwise, one may wonder this paper does not fall within the scope of the journal “administrative sciences”.
Author Response
Thank you for the expert feedback. Implications of ‘acceptance theory’ has been added to strengthen the theoretical model. Stratified sampling technique was used to ensure proportionate representation of different age groups, i.e. 50% of the population consists of below 45 years age group, and 50% of the population represents above 45 years age group. The age segregation is imperative in the sense that aged people in developing economies age are known for exhibiting lack of technology acceptance.
The article falls well within the scope of ‘administrative sciences’ journal, as it addresses the administrative implications of ERP and technology acceptance issues for the management. Unfortunately, we couldn’t find any relevant paper published in “administrative sciences”; our paper will somehow fill this gap.
Reviewer 2 Report
in my review of the first version of that paper, I addressed many topics in all sections of the paper. Especially, the literature is improved. The novelty and value of the paper is explained in detail. In addition I had serious concerns about the relevance of the following hypotheses: - H1: Individual impact has a positive relation with service quality in healthcare. - H2: Organizational impact has a positive relation with service quality in healthcare. - H3: Information quality impact has a positive relation with service quality in healthcare. - H4: System quality impact has a positive relation with service quality in healthcare. The authors have substantially reworked this analysis. In the present form I recommend to accept this paper.Author Response
Thank you for your motivational feedback.
Reviewer 3 Report
The authors have done a great job on the revision. Happy to accept.
Author Response
Thank you for your motivational feedback.
Round 2
Reviewer 1 Report
This paper has been revised following my comments and suggestions.